# Assessment of Agro-Morphologic Performance, Genetic Parameters and Clustering Pattern of Newly Developed Blast Resistant Rice Lines Tested in Four Environments

**Raieah Saiyedah Sabri [1], Mohd Y. Rafii [1,2,\*], Mohd Razi Ismail [1,2], Oladosu Yusuff [1], Samuel C. Chukwu [1,3] and Nor'Aishah Hasan [1,4]**

[1] Institute of Tropical Agriculture and Food Security, Universiti Putra Malaysia, Serdang, Selangor 43400, Malaysia; raieahsaiyedah@gmail.com (R.S.S.); razi@upm.edu.my (M.R.I.); oladosuy@gmail.com (O.Y.); chukwusamuel54@yahoo.com (S.C.C.); aishahnh@ns.uitm.edu.my (N.A.H.)

[2] Department of Crop Science, Faculty of Agriculture, Universiti Putra Malaysia, Serdang, Selangor 43400, Malaysia

[3] Department of Crop Production and Landscape Management, Ebonyi State University, Abakaliki PBM 053, Nigeria

[4] Faculty of Applied Science, Universiti Teknologi MARA, Kuala Pilah, Negeri Sembilan 72000, Malaysia

\* Correspondence: mrafii@upm.edu.my; Tel.: +60397691043

**Abstract:** Multi-environmental yield trial is very vital in assessing newly developed rice lines for its adaptability and stability across environments especially prior to release of the newly developed variety for commercial cultivation. The growth performance and phenotypic variability of these genotypes are the combination of environment, genotype and genotype by environment (G×E) interaction factors. Thus, evaluation creates an opportunity for effective selection of superior genotypes. The objectives of this study were to evaluate the newly developed blast resistant rice lines in varied environmental conditions, precisely measure the response of the advanced lines in multiple environments and classify the genotypes into groups that could serve as varieties for commercial cultivation. Genetic materials included 18 improved blast resistant rice lines and the recipient parent MR219. The total of 19 newly developed genotypes was evaluated under four varied environments in Peninsular Malaysia. The experiments were carried out using randomized complete block design (RCBD) with three replications at each environment. Data were collected on the vegetative, yield and yield component traits. Descriptive statistics (mean performance) and analysis of variance were conducted using SAS Software version 9.4. Genotypic and phenotypic coefficients, phenotypic variance component, heritability and genetic advance were also determined. Analysis of variance revealed that all traits were significantly different for genotypes except days to maturity, number of filled grains and total number of grains. Meanwhile, all the traits differed significantly for genotype × environment (G×E) except number of tillers per hill and number of panicles per hill. Low heritability (<30%) was found for all the traits. Similarly, low genetic advance was also observed for all the traits except for number of tillers per hill and number of panicles per hill. yield per hectare had significant and positive correlation with most evaluated traits except for days to flowering, days to maturity, plant height and number of unfilled grains. Cluster analysis classified the 19 evaluated genotypes into six groups. Therefore, the six clusters/groups of genotypes were recommended as varieties for commercial cultivation in Malaysia and other rice growing regions.

**Keywords:** rice genotypes; blast resistant genotype; genotypic coefficient of variation (GCV); phenotypic coefficient of variation (PCV); heritability values

## 1. Introduction

Rice (*Oryza sativa* L.) is a staple food for over three billion of the world's population—mostly in the Asian continent [1]. The present and future global food security on rice is important due to the amount of daily calories and protein intake derived from it [2]. Over the years, there is a continuous increase in human population and rice consumption. This has caused vast gap between rice demand and its production, thus creating a need to increase the current rice yield potential [3]. However, rice productivity and sustainability are constrained by biotic and abiotic stress which was further intensified by climate change, weather variability and water shortage [4]. In Malaysia, rice is completely guarded by subsidizing, price control and import tariffs because it is heavily influenced by social, economic and political sensitivity [5,6]. The current mean yield of Malaysian rice production was 4.03 t/ha with an annual production of 2.2 million tons [7]. This output can only meet 72.3% of its local demands [8]. Malaysia lacks comparative advantage in paddy production and limited available land area for rice production compared to bordering countries like Vietnam and Thailand. Consequently, the country needs to depend on imported rice to fulfill its level of self-sufficiency (SSL) [9].

One of the significant factors that is seriously threatening high production of rice is blast disease caused by the fungal pathogen *Magnaporthe oryzae* [10–12]. Blast disease accounted for 10%–20% yield loss in susceptible varieties and can be more than 80% in severe conditions [13]. Nasruddin and Amin [14] reported the use of integrated management strategy which includes resistant cultivar, suitable date of planting and fungicide to control blast disease. However, this disease occurs due to the interaction of a favorable environment, a susceptible genotype and a virulent pathogen. Hence, the utilization of resistant cultivar is a favorable way to reduce the use of destructive pesticides [15,16]. Molecular markers have been intensively used to pyramid beneficial and multiple alleles to develop new rice blast resistant varieties [17–19]. This approach is the most cost-effective and environmental safety in order to manage rice blast disease [20]. However, an evaluation on the performance of the improved genotypes is needed before the selection of ideal genotype. This is because the phenotypic performance and adaption to adverse environments are influenced by the genotype (G), environment (E) and genotype × environment (G×E) interaction [21,22]. Furthermore, variation in their performance is more significant due to higher proportion of this G×E interaction than genotypic main effects [23–25]. This G×E interaction is described as the differential response in respective genotype performance in dissimilar or unpredictable environments [26]. Genotype selection based on stability and adaptability in multiple environmental conditions is important before the recommendation for large cultivation [27–30]. Therefore, this study was conducted to evaluate the newly developed blast resistant rice lines in varied environmental conditions, precisely measure the response of these advanced lines in multiple environments and classify the genotypes into groups that could serve as varieties for commercial cultivation.

## 2. Materials and Methods

### 2.1. Planting Materials

Eighteen advance lines of $BC_2 F_2$ generation from backcross of MR219 and Pongsu Seribu 2 were used in this study. Introgression lines were developed from these parents (MR219 x Pongsu Seribu 2) using marker-assisted backcross selection (MABS). The donor parent, Pongsu Seribu 2 (PS2), developed by Malaysian Agricultural Research and Development Institute (MARDI) possesses broad-spectrum resistance against blast fungal isolates. MR219 has high yielding potential with a suitable grain quality and good eating quality. Unfortunately, this variety is susceptible to blast. The $F_1$ plant produced from the cross between MR219 and Pongsu Seribu 2 was later backcrossed with MR219 to produce $BC_1F_1$ and subsequently backcrossed again with MR219 to derive $BC_2F_1$. The $BC_2F_1$ was allowed to self to produce $BC_2 F_2$ population among which 18 advance lines with the highest genome recovery and phenotypic characteristics similar to the MR219 were selected. The 18 advance lines derived had advantage of blast resistance in the four environments as well as high yielding characteristic similar to

its recurrent parent MR219 [12]. The planting materials, i.e., the 18 advance breeding genotypes and MR219 were subjected to multi-environmental field trials.

## 2.2. Experimental Environments

The experiment was carried out between 2015 and 2017 in four different environments that represented the major rice growing areas in Peninsular Malaysia. The environment is referred to as the combination of year and location that covered a wide range of conditions such as differing temperatures (warm to moderate climate conditions), rainfall, water regimes (full and supplementary irrigation), soil types (loam and clay loam), biotic (pests and diseases), cropping seasons (main and off season) and conditions of management (research stations and farmers' fields). The four environments tested and their site description as well as planting period for each site is as presented (Table 1).

**Table 1.** Environmental data and description.

| Code | Planting Period | Location | Altitude (m) | Av. Temp. Min–Max (°C) | Av. Humidity (%) | Rainfall (Mean) |
|---|---|---|---|---|---|---|
| EN1 | September 2015–January 2016 | 3° 25′N 101° 10′E | 3 | 23–31 | 83 | 782.4 (195.6) |
| EN2 | February–June 2016 | 3° 25′N 101° 10′E | 3 | 25–37 | 65 | 482.7 (120.7) |
| EN3 | December 2016–March 2017 | 5° 59′N 100° 24′E | 18 | 25–38 | 63 | 486.9 (121.7) |
| EN4 | May–September 2017 | 3° 02′N 101° 42′E | 32 | 24–38 | 67 | 623.4 (115.9) |

Note: EN1—TanjungKarang; Selangor; EN2—TanjungKarang; Selangor; EN3—Kota SarangSemut; Kedah; EN4—Kota SarangSemut.

## 2.3. Experimental Scheme and Cultural Practices

Field experiment at each environment was laid out in a randomized complete block design (RCBD) in three replications with plot size of 16.25 by 4 m. The subplot size for each replication is 4 by 4 m with planting distance of 25 cm within and between rows. The 18 advance lines together with the MR219 control was first deactivated seed dormancy by oven drying at 40 °C for 24 h. The seeds were germinated by fully soaking in water overnight in a petri dish to induce pre-germination. Later, the water was removed, and the seeds were kept moist for three days. To avoid drying out during the three days, water was added to the seeds to keep it moist. After three days, the seeds were transferred to soil-filled plastic trays that were prepared earlier. The seedlings were allowed to grow for 21 days in the nursery before transplanting it to the field. A single seedling per hill at 21 days was manually transplanted to the rice field at each of the environment. All the cultural practices from land preparation until harvesting were done following MARDI recommendations. The field was irrigated with an average of 10 cm water above ground surface level throughout the experiments. Fertilization was applied following the recommendations from MARDI where 42 kg/ha muriate of potash and 57 kg/ha triple superphosphate were applied at day 15 after transplanting. Urea was applied at 35, 55 and 75 days after transplanting in splits at 80, 12 and 20 kg/ha, respectively. The insecticides; Malathion and Lambda-cyhalothrin were applied at the recommended rate of 20 mL per 15 L knapsack sprayer when needed. Regular, hand-weeding was done to remove narrow leaf weeds. In the case of broad leaf weeds, halosulfuron-methyl was applied at the rate of 40 mL per 15 L sprayer.

## 2.4. Data Collection

Data were collected on 13 quantitative traits, viz; number of days to flowering (DTF) and maturity (DTM), plant height (PH cm), number of tillers per hill (TPH) and panicles per hill (PPH), panicle length (PL cm), number of filled grains (FG) and unfilled grains (UFG) per panicle, total number of

grains per panicle (TG), percentage of filled grains (PFG), 1000-grain weight (THW g), total weight of grains per hill (TW g) and yield (YLD t/ha). Sampling was conducted on five plants for each genotype from each replication following the IRRI [31] procedure.

*2.5. Data Analysis*

Analysis of variance (ANOVA) was done for all the traits using the SAS program version 9.4 (SAS Institute, Cary, NC, USA) to determine the variation among the genotypes, environment and genotypes by environment. In addition, descriptive statistics such as mean, range, standard deviation and coefficient of variation (CV) were calculated for each trait. Mean comparisons were performed using Tukey's test. Correlation coefficients were analyzed using SAS Software (version 9.4) to study the relationship between traits. Multivariate analysis was done using SAS software to determine the pattern of interaction between the genotype and environment. The method used in this study was cluster analysis. The Euclidean distances amidst the 19 improved genotypes evaluated were observed by using the standardized morphologic data to construct an UPGMA dendrogram. The variance components were also determined from the expected mean square using proc varcomp with restricted maximum likelihood (REML) method in SAS [10,27]. However, phenotypic variance was calculated using the formula (Equation (1)):

$$\sigma^2_p = \sigma^2_g + \sigma^2_{ge} + \sigma^2_\epsilon \tag{1}$$

where, $\sigma^2_p$=Phenotypic variance, $\sigma^2_g$ = genotypic variance, $\sigma^2_{ge}$ =variance of G×E and $\sigma^2_\epsilon$ = error variance.

The percentage of GCV and PCV values were classified as low if the range is between 0% and 10%, 10%–20% as moderate and 20% and above as high according to Sivasubramaniam & Madhava [32]. Classification of heritability percentage was done according to Falconer [21] that categorized it into three; low (0%–30%), moderate (30%–60%) and high (≥ 60%). Johnson et al. [33] classified the percentage of genetic advance into low when the value is 0%–10%, moderate with value of 10%–20% and high when the value is more than 20%. The estimation of genetic parameters such as variance components, broad sense heritability and expected genetic advance was done as follows:

| | | | |
|---|---|---|---|
| i. | phenotypic coefficient of variation (PCV) | $PCV = \dfrac{\sqrt{\sigma^2 p}}{X} \times 100$ | Where: $\sigma^2 p$ = phenotypic variance; X= mean of the trait |
| ii. | genotypic coefficient of variation (GCV) | $GCV = \dfrac{\sqrt{\sigma^2 g}}{X} \times 100$ | Where; $\sigma^2 g$ = genotypic variance; X = mean of the trait |
| iii. | broad sense heritability | $h^2B = \dfrac{\sigma^2 g}{\sigma^2 p}$ | Where: $\sigma^2 g$ = genotypic variance; $\sigma^2 p$ = phenotypic variance |
| iv. | Expected genetic advance | $GA = K \times \sqrt{\sigma^2 p} \times h^2B$ | Where: K = constant that represents the selection intensity (when k is 5% the value is 2.06); $\sqrt{\sigma^2 p}$= standard deviation of phenotypic variance; $h^2B$ = heritability in a broad sense |

## 3. Results

*3.1. Agro-Morphologic Traits, Genotype and G×E Interactions*

The combined analysis of variance and mean comparison for 13 agro-morphologic traits evaluated in four different environments for varietal assessment are as presented in Tables 2 and 3, respectively. days to flowering showed a significant difference ($p \leq 0.05$) among genotypes while highly significant

differences ($p \leq 0.01$) were observed for G×E. Environment accounted for the highest percentage of variation of 75%, while genotypes and G×E accounted for 8.9% and 4.89% variations, respectively (Table 2). The combined analysis showed that number of days to flowering started 70.92 days after transplanting in G7 followed by G6, G10 and G15 at 71.67, 71.17 and 71.58 days, respectively. For days to maturity, no significant difference was observed for genotypes while significant ($p \leq 0.01$) G×E interaction was observed. total number of grains had highly significant difference ($p \leq 0.01$) for G×E with 14.54% of variation. Genotypes had no significant difference in total number of grains but recorded 12.10% of variation. The genotypes varied significantly ($p \leq 0.05$) in percentage of filled grains with 1.66% total variation. A high significant difference was observed for G×E with total of 0.90% variation. Highest percentage of filled grains was observed in G18 with 83.89%. thousand-grain weight recorded a significant difference for genotypes while highly significant difference was observed in G×E. Genotypes and G×E recorded 17.66% and 8.71% variation, respectively. Thousand grain weights showed all genotypes had weight of more than 21 g with an average of all genotypes at 23.31 g. The highest thousand-grain weight was recorded in G18 with 24.78 g. This genotype had nearly the same weight with G4 (24.19 g), G14 (24.04 g) and MR219 (24.42 g). Meanwhile, total grains weight revealed a high significant difference ($p \leq 0.01$) for genotypes and a significant difference ($p \leq 0.05$) for G×E. Percentage of variation was counted at 23.09% for genotypes and 8.38% for G×E. The mean of total grains weight per hill for all genotypes was 37.85 g (Table 3). Genotype G18 had highest total grains weight with 54.23 g. High significant difference ($p \leq 0.01$) for genotypes and a significant difference ($p \leq 0.05$) for G×E were observed for yield per hectare. Variation at 23.10% for genotypes and 8.38% for G×E were recorded for yield per hectare trait. The highest yield per hectare in tons (t) was recorded in G18 with the value of 8.68 t/ha. It was followed by MR219 with 7.33 t/ha and G17 with 7.03 t/ha.

### 3.2. Genotypic and Phenotypic Coefficient of Variability (GCV and PCV)

The level of variability among the 13 agro-morphologic traits with respect to variance components was presented in Table 4. The GCV values ranged from 0.61–11.91%. Low GCV value was recorded for all traits, having value less than 10% except for the number of tillers per hill, number of panicles per hill, number of unfilled grains, total grains weight and yield per hectare. These five traits had a moderate percentage of GCV (10–20%). As for PCV, number of tillers per hill, number of panicles per hill, number of unfilled grains, total grains weight and yield per hectare recorded a high percentage of PCV (>20%). However, moderate value was recorded for the number of filled grains and total number of grains (10–20%). Low PCV value was observed in other traits (<10%).

### 3.3. Heritability and Genetic Advance

The result in Table 4 shows the broad-sense heritability and genetic advance of the evaluated traits. Low heritability value was recorded for all 13 traits which range between 2.42% and 27.08%. The highest heritability value was found in number of panicles per hill (27.08%) followed by panicle length (26.42%) and number of tillers per hill (26.11%). In this study, GA value of these 13 traits ranged between 0.24–12.40%. Moderate genetic advance (GA) was found in number of tillers (12.05%) and number of panicles (12.40%). Other 11 traits showed low GA percentage.

### 3.4. Correlation and Cluster Analyses

Besides days to 50% flowering and number of unfilled grains per hill, all other ten traits showed a positive relationship with the final yield per hectare (Table 5). Highly significant correlation coefficients with yield were discovered in nine traits. days to maturity correlated positively with days to flowering, number of tillers per hill, number of panicles per hill, filled grain, percentage filled grain and thousand grain weight, but had negative association with plant height and unfilled grain. number of tillers per hill had a significant positive correlation with number of panicles per hill, filled grain, percentage filled grains, total weight and yield. panicle length also had a significant positive correlation with unfilled

grain, total number of grains, thousand grain weight, total weight and yield, but negatively correlated with percentage filled grain. Filled grain negatively correlated with unfilled grain and thousand grain weight but had a significant positive correlation with total number of grains, percentage filled grains, total weight and yield. The cluster analysis classified the genotypes into six major groups at 0.19 dissimilarity coefficients (Figure 1). This revealed the effectiveness of quantitative/morphologic traits in grouping the rice genotypes. It also showed a high level of morphologic variations present among the assessed genotypes. The largest group was Group I that consist of eight genotypes which are G1, G5, G4, G12, G8, G14, G6 and G10. This was followed by Group V with five genotypes (G2, G3, G13, G15 and G7) and Group III with three genotypes (G17, G18 and MR219). Group II, IV and VI only consist of one genotype in each group (Figure 1).

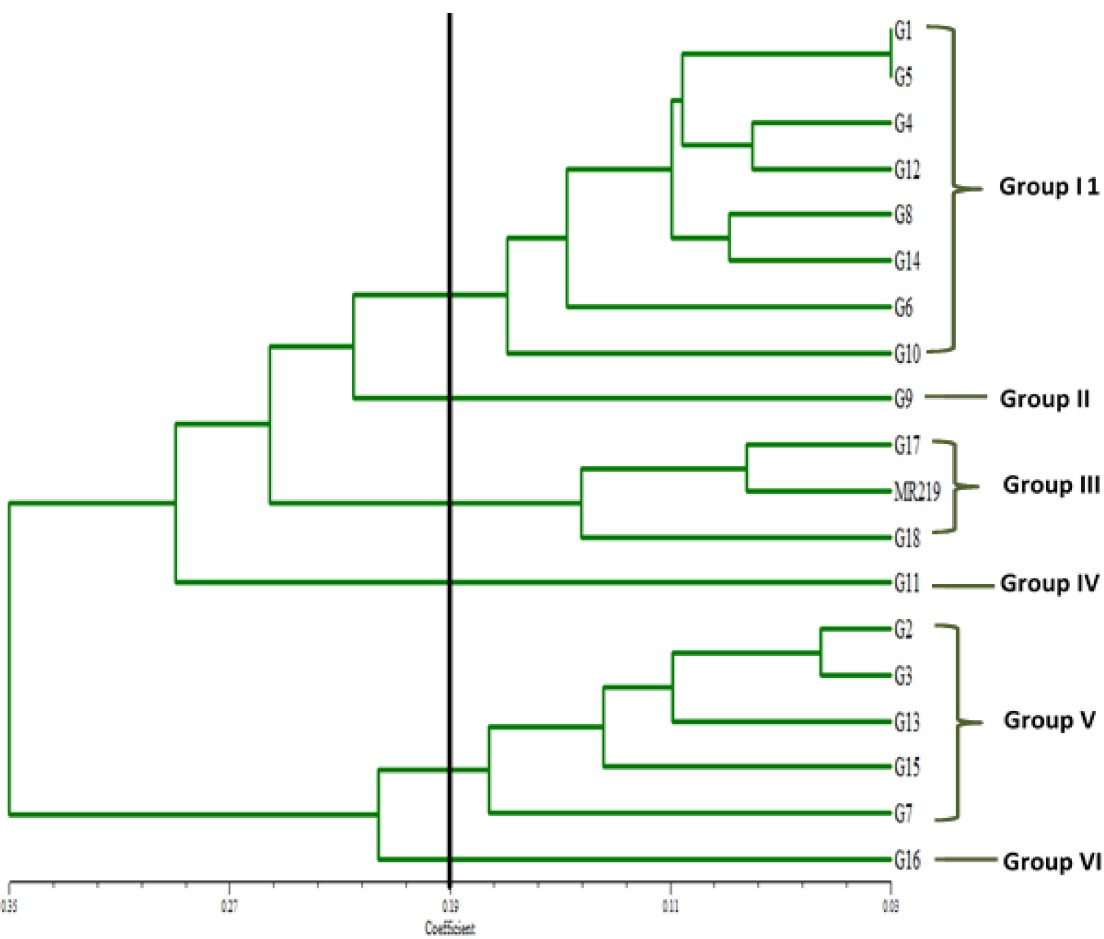

**Figure 1.** Clustering pattern of 19 rice genotypes evaluated in four different environments.

**Table 2.** mean square of combined analysis of variance for 13 traits assessed in four environments.

|  | SOV | Blocks (Environment) | Genotypes (G) | Environments (S) | G×S | Error |
|---|---|---|---|---|---|---|
|  | DF | 8 | 18 | 3 | 54 | 144 |
| DTF | MS | 18.34** | 17.78* | 149.97** | 9.77** | 3.94 |
|  | TSS (%) | 9.18 | 8.9 | 75.06 | 4.89 | 1.97 |
| DTM | MS | 133.93** | 19.96$^{ns}$ | 856.91** | 15.20** | 8.41 |
|  | TSS (%) | 12.95 | 1.93 | 82.84 | 1.47 | 0.81 |
| PH | MS | 127.44** | 60.67** | 12,680.89** | 22.82** | 12.11 |
|  | TSS (%) | 0.99 | 0.47 | 98.27 | 0.18 | 0.09 |
| NTH | MS | 95.95** | 105.02** | 726.42** | 11.81$^{ns}$ | 23.13 |
|  | TSS (%) | 9.97 | 10.91 | 75.49 | 1.23 | 2.4 |
| NPH | MS | 68.43** | 78.49** | 478.86** | 12.40$^{ns}$ | 15.13 |
|  | TSS (%) | 10.47 | 12.01 | 73.3 | 1.9 | 2.32 |
| PL | MS | 1.54** | 4.67** | 97.22** | 1.29** | 0.53 |
|  | TSS (%) | 1.46 | 4.44 | 92.37 | 1.23 | 0.5 |
| FG | MS | 1791.21** | 2877.37$^{ns}$ | 38,887.19** | 1743.21** | 608.58 |
|  | TSS (%) | 3.9 | 6.27 | 84.71 | 3.8 | 1.32 |
| UFG | MS | 1979.64** | 700.88* | 45,613.77** | 352.45** | 212.92 |
|  | TSS (%) | 4.05 | 1.43 | 93.36 | 0.72 | 0.44 |
| TG | MS | 1384.19* | 1595.14$^{ns}$ | 5975.58** | 1327.15** | 686.72 |
|  | TSS (%) | 12.62 | 14.54 | 54.48 | 12.1 | 6.26 |
| PFG | MS | 280.32** | 145.62* | 8234.20** | 78.53** | 34.06 |
|  | TSS (%) | 3.2 | 1.66 | 93.86 | 0.9 | 0.38 |
| TGW | MS | 10.46** | 6.97* | 17.05** | 3.44** | 1.56 |
|  | TSS (%) | 26.49 | 17.66 | 43.19 | 8.71 | 3.95 |
| TW | MS | 449.08** | 380.11** | 594.18** | 138.06** | 85.08 |
|  | TSS (%) | 27.27 | 23.09 | 36.09 | 8.38 | 5.17 |
| YLD | MS | 11.49** | 9.73** | 15.19** | 3.53** | 2.18 |
|  | TSS (%) | 27.28 | 23.1 | 36.06 | 8.38 | 5.18 |

Note: * significant at $p \leq 0.05$, ** highly significant at $p \leq 0.01$, ns: not significant at $p > 0.05$; SOV—source of variation; DF—degrees of freedom; MS—mean square; TSS—total sum of square; DTF—days to flowering; DTM—days to maturity; PH—plant height; NTH—number of tillers per hill; NPH—number of panicles per hill; PL—panicles length; FG—number of filled grains per panicle; UFG—number of unfilled grains per panicle; TG—total number of grains; PFG—percentage of filled grains; TGW—thousand-grain weight; TW—total weight; YLD—yield per hectare.

**Table 3.** Mean for 13 traits of 19 rice genotypes tested in four environments.

| Genotypes | DTF (Day) | DTM (Day) | PH (cm) | NTH (no) | NPH (no) | PL (cm) | FG (no) | UFG (no) | TG (no) | PFG (%) | TGW (g) | TW (g) | YLD(t/ha) |
|---|---|---|---|---|---|---|---|---|---|---|---|---|---|
| G1 | 72.25 ab | 104.25 | 98.52 ab | 19.92 de | 17.00 de | 24.30 abc | 192.33 | 47.91 | 240.25 | 79.35 | 23.21 ab | 38.60 ab | 6.17 ab |
| G2 | 72.00 ab | 103.17 | 100.72 ab | 19.83 e | 17.50 de | 24.81abc | 178.67 | 55.41 | 234.08 | 77.29 | 22.89 ab | 33.05b | 5.29 b |
| G3 | 73.50 ab | 105.33 | 100.93 ab | 22.17 bcde | 18.58 cde | 24.85abc | 176.83 | 55.75 | 232.59 | 76.06 | 23.93 ab | 34.30 b | 5.49 b |
| G4 | 72.50 ab | 104.75 | 100.12 ab | 22.25 bcde | 18.25 cde | 24.86abc | 186.58 | 48.25 | 234.83 | 78.39 | 24.19 ab | 33.83 b | 5.41 b |
| G5 | 73.00 ab | 105 | 99.32 ab | 20.75 cde | 18.67 cde | 24.38abc | 194.25 | 49.33 | 243.58 | 79.97 | 23.85 ab | 39.95 ab | 6.39 ab |
| G6 | 71.67 ab | 104.17 | 101.73 a | 19.75 e | 16.58 e | 25.15 a | 193.42 | 40.75 | 234.17 | 81.91 | 23.22 ab | 37.80 ab | 6.05 ab |
| G7 | 70.92 b | 101 | 99.55 ab | 22.75bcde | 19.83bcde | 24.53abc | 165.92 | 64.33 | 230.25 | 72.49 | 23.65 ab | 33.27 b | 5.32 b |
| G8 | 72.33 ab | 103.83 | 98.93 ab | 23.17bcde | 20.25bcde | 24.67abc | 185.08 | 44.75 | 229.83 | 80.13 | 23.07 ab | 38.12 ab | 6.10 ab |
| G9 | 74.25 ab | 106.17 | 97.25 ab | 25.17 ab | 20.75abcde | 23.36abc | 169.42 | 48.17 | 217.59 | 79.14 | 22.54 ab | 35.16 b | 5.63 b |
| G10 | 71.17 b | 102.75 | 94.35 b | 22.08bcde | 19.83bcde | 23.53abc | 195.33 | 51.92 | 247.25 | 79.71 | 22.75 ab | 36.82 ab | 5.89 ab |
| G11 | 73.92 ab | 105.58 | 98.61 ab | 22.83bcde | 19.92bcde | 25.15 a | 226.58 | 43.33 | 269.92 | 82.92 | 22.49 ab | 41.00 ab | 6.56 ab |
| G12 | 75.92 a | 106.25 | 98.05 ab | 21.25 cde | 18.25 cde | 24.36abc | 191.92 | 46.5 | 238.42 | 80.47 | 23.00 ab | 34.21b | 5.47 b |
| G13 | 73.25 ab | 103.83 | 99.04 ab | 22.83bcde | 19.00 cde | 23.37bc | 175.17 | 60.5 | 235.67 | 74.42 | 23.06 ab | 33.44 b | 5.35 b |
| G14 | 72.00 ab | 103.58 | 94.57 ab | 24.67abcde | 21.33abcde | 23.48abc | 178.67 | 47.08 | 225.75 | 79.23 | 24.04 ab | 40.91 ab | 6.55 ab |
| G15 | 71.58 ab | 103.58 | 94.28 b | 26.92 ab | 23.25 abc | 23.17 c | 170.58 | 56.42 | 227 | 74.75 | 21.63 b | 33.76 b | 5.40 b |
| G16 | 73.67 ab | 105.17 | 97.12 ab | 21.17 cde | 17.67 de | 24.48abc | 156.92 | 58.59 | 215.5 | 72.97 | 23.01 ab | 31.01 b | 4.96 b |
| G17 | 73.83 ab | 105.33 | 98.79 ab | 25.08 abcd | 22.08abcd | 24.62abc | 194.42 | 39.25 | 233.66 | 83.61 | 23.08 ab | 43.95 ab | 7.03 ab |
| G18 | 73.25 ab | 104.75 | 100.48 ab | 29.58 a | 25.00 ab | 25.06 ab | 199.92 | 38.33 | 238.25 | 83.89 | 24.78 a | 54.23 a | 8.68 a |
| MR219 | 72.83 ab | 105.5 | 101.10 ab | 29.67 a | 25.67 a | 24.44abc | 193.17 | 38.83 | 232 | 82.83 | 24.42 ab | 45.79 ab | 7.33 ab |
| Mean | 72.83 | 104.42 | 98.6 | 23.25 | 19.97 | 24.36 | 185.53 | 49.23 | 234.77 | 78.92 | 23.31 | 37.85 | 6.06 |
| HSD($p$ = 0.05) | 4.31 | 5.88 | 7.2 | 5.18 | 5.31 | 1.71 | 62.97 | 28.31 | 54.94 | 13.37 | 2.8 | 17.72 | 2.83 |
| CV | 4.03 | 3.9 | 5 | 26.93 | 23.76 | 4.34 | 20.54 | 43.32 | 13.94 | 11.98 | 7.73 | 32.64 | 32.64 |
| Max | 77.33 | 112.67 | 111.73 | 49 | 30.33 | 26.7 | 275 | 132.33 | 315.33 | 90.97 | 28.36 | 70.17 | 11.23 |
| Min | 65.33 | 95.67 | 88.13 | 12.67 | 10.33 | 22.07 | 108 | 25.33 | 169.33 | 46.92 | 18.32 | 13.78 | 2.2 |

Note: HSD—honestly significant difference by Tukey's test (means within each column with same letter are not significantly different with HSD test *p* > 0.05), CV—coefficient of variation, DTF—days to flowering, DTM—days to maturity, PH—plant height, NTH—number of tillers per hill, NPH—number of panicles per hill, PL—panicles length, FG—number of filled grains per panicle, UFG—number of unfilled grains per panicle, TG—total number of grains per panicle, PFG—percentage of fertile grains, TGW—thousand-grain weight; TG—total grains per hill weight; YLD—yield per hectare.:

**Table 4.** Variance components, coefficients of variation, heritability and genetic advance of the rice genotypes for 13 traits assessed in four environments.

| SOV | DTF (Day) | DTM (Day) | PH (cm) | NTH (No) | NPH (No) | PL (cm) | FG (No) | UFG (No) | TG (No) | PFG (%) | TGW (g) | TW (g) | YLD (t/ha) |
|---|---|---|---|---|---|---|---|---|---|---|---|---|---|
| $\sigma^2_g$ | 0.67 | 0.40 | 3.15 | 7.08 | 5.34 | 0.28 | 94.51 | 29.04 | 22.33 | 5.59 | 0.29 | 20.17 | 0.52 |
| $\sigma^2_{gs}$ | 1.94 | 2.26 | 3.57 | 0.00 | 0.00 | 0.25 | 378.21 | 46.51 | 213.48 | 14.82 | 0.63 | 17.66 | 0.45 |
| $\sigma^2_e$ | 3.94 | 8.41 | 12.11 | 20.04 | 14.38 | 0.53 | 608.58 | 212.92 | 686.72 | 34.06 | 1.56 | 85.08 | 2.18 |
| $\sigma^2_p$ | 6.55 | 11.07 | 18.83 | 27.12 | 19.72 | 1.06 | 1081.30 | 288.47 | 922.53 | 54.47 | 2.48 | 122.91 | 3.15 |
| Mean | 72.83 | 104.42 | 98.60 | 23.25 | 19.97 | 24.36 | 185.54 | 49.23 | 234.77 | 78.92 | 23.31 | 37.85 | 6.06 |
| $h^2_B$ (%) | 10.23 | 3.61 | 16.72 | 26.11 | 27.08 | 26.42 | 8.74 | 10.07 | 2.42 | 10.26 | 11.70 | 16.41 | 16.51 |
| GCV (%) | 1.12 | 0.61 | 1.80 | 11.44 | 11.57 | 2.17 | 5.24 | 10.95 | 2.01 | 3.00 | 2.31 | 11.87 | 11.91 |
| PCV (%) | 3.51 | 3.19 | 4.40 | 22.40 | 22.24 | 4.23 | 17.72 | 34.50 | 12.94 | 9.35 | 6.76 | 29.29 | 29.30 |
| GA (%) | 0.74 | 0.24 | 1.52 | 12.05 | 12.40 | 2.30 | 3.19 | 7.15 | 0.65 | 1.98 | 1.63 | 9.90 | 9.97 |

Note: SOV—source of variation; DF—degrees of freedom; $\sigma^2_g$—variance of genotypic; $\sigma^2_{gs}$—variance of genotypic by environment; $\sigma^2_e$—variance of error; $\sigma^2_p$—variance of phenotypic; $h^2_B$—broad-sense heritability; PCV—phenotypic coefficient of variation; GCV—genotypic coefficient of variation; GA—genetic advance; DTF—days to flowering; DTM—days to maturity; PH—plant height; NTH—number of tillers per hill; NPH—number of panicles per hill; PL—panicle length; FG—number of filled grains per panicle; UFG—number of unfilled grains per panicle; TG—total number of grains per panicle; PFG—percentage of filled grains per panicle; TGW—thousand-grain weight; TW—total grains weight per hill; YLD—yield per hectare.

**Table 5.** Correlation coefficients among the quantitative traits evaluated.

| | DTF | DTM | PH | NTH | NPH | PL | FG | UFG | TG | PFG | TGW | TW | YLD |
|---|---|---|---|---|---|---|---|---|---|---|---|---|---|
| DTF | | 0.43** | -0.30** | 0.183** | -0.02 | -0.14* | 0.20** | -0.19** | 0.06 | 0.22** | -0.12 | -0.07 | -0.07 |
| DTM | | | -0.30** | 0.28** | 0.42** | -0.06 | 0.30** | -0.27** | 0.12 | 0.31** | 0.13* | 0.122 | 0.128 |
| PH | | | | -0.49** | -0.27** | 0.72** | -0.51** | 0.71** | 0.05 | -0.73** | 0.25** | 0.02 | 0.02 |
| NTH | | | | | 0.80** | -0.31** | 0.36** | -0.41** | 0.05 | 0.45** | -0.03 | 0.38** | 0.38** |
| NPH | | | | | | -0.14* | 0.32** | -0.28** | 0.14* | 0.33** | 0.11 | 0.45** | 0.45** |
| PL | | | | | | | -0.16** | 0.55** | 0.33** | -0.49** | 0.26** | 0.22** | 0.22** |
| FG | | | | | | | | -0.63** | 0.65** | 0.79** | -0.14* | 0.46** | 0.46** |
| UFG | | | | | | | | | 0.18** | -0.96** | 0 | -0.20** | -0.20** |
| TG | | | | | | | | | | 0.06 | -0.17** | 0.39** | 0.38** |
| PFG | | | | | | | | | | | -0.04 | 0.31** | 0.31** |
| THW | | | | | | | | | | | | 0.22** | 0.22** |
| TW | | | | | | | | | | | | | 1** |
| YLD | | | | | | | | | | | | | |

Note: *significant at 0.05 probability level, **highly significant at 0.01 probability level, DTF—days to flowering; DTM—days to maturity; PH—plant height; NTH—number of tillers per hill; NPH—number of panicle per hill; PL—panicle length; FG—number of filled grain; UFG—number of unfilled grain; TG—total number of grain per panicle; PFG—percentage of fertile grain; THW—thousand-grain weight; TW—total weight per hill; YLD—yield in t/ha.

## 4. Discussion

The number of tillers in rice is a significant agronomic trait that directly affects grain yield due to its positive relationship with the production of panicle that will bear the rice grains. The presence of fewer tillers produced fewer panicles, while excess tillers caused higher tillers abortions, small panicles and poor grain filling [34,35]. Ranawake and Amarasinghe [36] reported the positive relationship of tillers and panicles that caused a decrease in grain output. In terms of individual tiller development, a water deficit at these reproductive stages causes irreparable loss of potential yield. Sometimes, several tillers had two panicles where the original panicle is sterile, but the second panicle became nonfunctional. This panicle originated from the flag leaf node that was developed after the original assimilate sink. In this study, the number of tillers produced was between 19 and 29, while the number of panicles ranged from 16 to 25. These are moderate amount of tillers and panicles in rice production. Rice yields are influenced by many factors, yet number of tillers and panicles are mainly considered as key factors.

Panicle length determines the amount of spikelet and grain that can be produced in a panicle. With good environmental conditions, longer panicle length gives room for more production of spikelet and total grain, resulting in high final yield output [37]. However,, high number of total grains is meaningless if the percentage of fertile grains is low. Percentage of fertile grains is the calculation on amount of filled grains over total grains in a panicle. High percentage as recorded in this study signifies high number of fertile grains on the panicle and it is preferred by farmers since it will directly influence the final yield. As revealed by this study, number of filled grains and percentage of fertile grains were positively correlated with the yield. All genotypes evaluated recorded high number of filled grains with average of more than 150 grains. In addition, number of unfilled grains produced was recorded at low levels (38–64 grains), hence, signify higher yield of rice being produced. However, if there were limitation or ineffective translocation of nutrition from the sources, it will affect the condition of the plant parts together with the rice grain. This results in poor production of rice grains [38]; a smaller number of total grains, small size, half-filled or unfilled grains. Although there is a high production of total grains, total grain weight is an important determinant of yield in rice. The study recorded moderate (31.01 g) and high (54.23 g) total grains weight. A decrease in grain weight is expected when there is water deficit in rice during grain filling. If the current photosynthate supply is limited, the ability to mobilize and translocate reserves would be adaptive. Reduction of grain weight may be due to defective grains. This can be signified by the value of 1000-grain weight. The low weight of 1000-grain indicates grains appearance; slender, small and thick hulls. In this work, moderate and high value of 1000-grain weight was observed that revealed the good appearance of rice grains; big and fully filled (fat). Therefore, final yield also was observed with the same results.

This study revealed wide phenotypic variability for 13 traits evaluated among the improved genotypes. The result indicates the existence of sufficient variations among the assessed genotypes for those traits under consideration that can help breeders in selection of ideal genotypes. These variations of the improved lines in relation to their agro-morphologic traits may be due to the fact that these lines were evaluated in environments that differed in temperature, humidity, rainfall and soil type. This signifies the need for multi-environmental trials of rice at various locations or environments in order to see how the genotypes react in different environments due to the presence of G×E interaction. Such variations in relation to the environment that influenced rice growth performance evaluation studies have extensively been reported previously [39–41].

Several reports have also been published on significant phenotypic variation among rice accessions although evaluated in only one environment [42–44]. Islam et al. [45] observed highly significant genetic variability in 113 rice genotypes for 18 traits studied. Newest study by Hosagoudar et al. [46] reported that there were significant differences that existed for traits investigated in 18 genotypes evaluated under hilly conditions. In Malaysia area of rice study, Tuhina-Khatun et al. [47] showed a significant existence of diversity level on 43 upland rice genotypes for 22 traits evaluated. Previously, Sohrabi et al. [48] presented significant differences in 50 Malaysian upland rice germplasms evaluated

for 12 traits. A significant variation, especially for G×E, signifies that the evaluated genotypes do react differently in a different environment. This fact shows that G×E interaction greatly gives influence in the selection process of superior genotype for release and this constraint needs to be solved by rice breeders

The study showed that the first environment (Tanjung Karang) planted from September 2015–January 2016 had the highest average relative humidity and rainfall although on a lower altitude of 3 m. There is no doubt that the period falls within the Malaysian heavy rain period (October–January). However, the rainfall regime and other environmental factors recorded in the other three environments were not the same. This corresponds with the significant difference observed in environment for most traits. The coefficient of variation in this study showed that PCV was relatively higher than GCV for all traits. The magnitude of differences between the value of PCV and GCV demonstrated how much environment influences the trait, where big differences referred to large environmental effect and little differences showed high genetic influence (27,39). The low differences between GCV and PCV obtained from this study implied that these traits can be used as selection criteria for further crop improvement since the variation in observed variables were mostly due to genetic factors. Previous study by Osman et al. [49] and Habib et al. [50] also found the same outcome for these traits. Anis et al. [51] and Nishanth et al. [52] also showed a difference of GCV and PCV values for several traits evaluated in 20 rice varieties and 525 germplasm lines for submergence tolerance, respectively. However, other seven traits showed high differences between PCV and GCV especially number of unfilled grains, total grain weight and yield per hectare. According to Hosagoudar and Kovi [53], the yield and yield component traits had large differences of PCV and GCV values among the 15 advanced rice genotypes evaluated for leaf blast reaction, genotypic performance and correlations. This big difference indicated large contribution of environmental factors on the phenotypic expression, hence, selection referring to these traits will not be much effective (27).

High value of GCV also indicates the existence of high genetic variation and selection using these traits to improve the genotypes could be effective. The results showed moderate value of GCV in number of tillers, number of panicles, number of unfilled grains, total grains weight and grain yield indicating moderate variability. This study was in agreement with Hasan et al. [54] who recorded moderate GCV in number of tillers, number of panicles and yield of newly developed blast resistant lines derived from crossing between MR263 and Pongsu Seribu 1. Srujuna et al. [55] also reported moderate GCV percentage for plant height, number of tillers and number of panicles on 29 evaluated rice genotypes. Value of GCV gives details on the genetic variability existing in quantitative traits, but the amount of variation that was heritable from the value of GCV is impossible to be determined. Thus, the amount of advance to be anticipated from selection is the best being visualized when using GCV value with the help of heritability estimates [56].

There were low heritability values for all traits evaluated in this study. This is an indication that direct selection using these traits would be ineffective due to high influence of environment. The results obtained in GA from this study were similar to the report by Kole et al. [57] for traits such as days to flowering, plant height and panicle length. Akinwale et al. [58] also reported days to maturity, panicle length, number of tillers and 1000-grain weight with low GA value. In contrast, Immanuel et al. [59] found high heritability value of more than 90% and high GA for most traits except for days to maturity and panicle length which had a moderate genetic advance. Work of Govintharaj et al. [60] also observed high and moderate heritability for several traits in the segregating population of blast introgressed lines.

In this study, the result showed that except days to flowering and number of unfilled grains per panicle, all other traits revealed positive relationship with yield per hectare. The association was highly significant with all traits studied except for days to maturity and plant height that were not significant with yield. The positive and significant correlation coefficients recorded between yield per hectare and other 10 quantitative traits indicated that such traits influenced the final yield of rice. These traits were suitable to be considered as yield prediction factors and deserved considerable importance during

selection in further study [39,61]. Manipulation of these traits may be useful to develop high yielding genotypes with desired traits. High yield was the results from the contribution of high number of tillers and number of panicles that helped in producing high number of total grains. Furthermore, higher yield was produced when there was high number of filled grains than unfilled grains. Number of tillers is a significant agronomic trait that directly affects grain yield due to its positive relationship with the production of panicle that will bear the rice grains. The presence of fewer tillers produced fewer panicles, while, excess tillers caused higher tiller abortions, small panicles and poor grain filling [34,35,37]. Ranawake and Amarasinghe [36] showed positive relationship between tillers and panicles that resulted in a decrease in grain output. The number of tillers and panicles produced in this study were moderate. In addition, longer panicle produces more grain yield. Few studies had been done that reported positive relationship between panicle length and yield [62–64].

This study revealed the effectiveness of quantitative or morphologic traits in grouping the rice genotypes. Group III clustered the genotypes that had high yield per hectare with good performances of other traits. It revealed that not only G17 and G18 have high yield, but also their morphologic performances were almost the same as the commercialized rice variety, MR219. Consequently, these genotypes could be fully utilized in a further breeding program for genotypes improvement. Ahmadikhah et al. [65] clustered 58 inbred rice lines into three distinct classes using 18 morphologic traits. The level of genetic diversity in crops is an important matter in maintaining and increasing agricultural productivity and it needs considerable attention.

## 5. Conclusions

Genotypes G17 and G18 maintained superiority in yield across the four environments and this showed that they have potentials for selection in further varietal improvement. Consequently, these genotypes can be fully utilized in a further breeding program for genotypes improvement. In particular, such genotypes would be useful to breeders in selecting high-yielding blast resistant rice varieties for different environments. The wide variability obtained among the 19 genotypes evaluated across the four varied environments is a cornerstone for plant breeding and the genotypes could serve as classical model for crop improvement. There was a high influence of environment in the genotypes' performance. However, the GxE result obtained for yield components such as number of tillers and number of panicles per hill revealed the existence of genetic variation at a considerable amount. The positive and significant correlation of these traits with yield showed that an increase in selection pressure on such traits could improve their agronomic yield. The new blast disease-resistant rice lines have great potentials in high yielding, blast resistance and useful for further crop improvement. Therefore, the six clusters/groups of genotypes obtained from this study are recommended as varieties for commercial cultivation in Malaysia and other rice growing regions.

**Author Contributions:** Conceived and designed the experiments: M.Y.R., R.S.S and M.R.I.; Performed the experiment: R.S.S. and M.Y.R. Analyzed the data: R.S.S., M.Y.R. and O.Y. Wrote and revised the article: R.S.S., M.Y.R., Y.O., S.C.C. and N.H., All authors have read and agreed to the published version of the manuscript.

**Funding:** The authors are grateful to the Ministry of Education Malaysia, for adequate research Higher Institution Centers of Excellence (HICoE) funding to conduct research on improvement of crop varieties for adaption to biotic and abiotic stress with grant number 6369105.

**Conflicts of Interest:** The authors declare no conflicts of interest.

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
