# Peer review of "Assessment of Agro-Morphologic Performance, Genetic Parameters and Clustering Pattern of Newly Developed Blast Resistant Rice Lines Tested in Four Environments"

_agronomy, doi:10.3390/agronomy10081098_

Round 1

Reviewer 1 Report

This paper is one of the many relevant studies looking at genotype x environment interaction. It has a high valorization potential as it (as also stated by the authors) should allow breeders to do more targeted selection of blast-resistant rice varieties which perform well (high yield) in different environments.

However, there are a number of concerns:

  • description of the experiment should improve: it is e.g. not clear whether crosses were backcrossed with MR219 or with Pongsu Seribu 2; also the agronomic (authors call them 'cultural') practices should be described more in detail (e.g. fertilizer and pesticide type and dosage); number the formula (of PVC, GVC, etc.); discuss the correlation and cluster analysis in the Material and Methods section, not in the results; 
  • data on 13 quantitative rice traits were collected, that were not independent, as clearly shown from the significance of the correlation coefficients. Principle Component Analysis (PCA) could have been performed prior to cluster analysis, which might have given more insight in the phenotypic variability;
  • Cluster Group III appears to contain the best (highest-yielding) rice varieties, but authors do not elaborate / conclude on how this finding helps breeders in selecting high-yielding blast resisting rice varieties for different environments. What other outstanding traits do genotypes in Group III have that might assist breeders in selecting such varieties?
  • § 3.1. is a written description of what can be found in tables 1 & 2. Authors should find a way to shorten it;
  • Summarize all data from § 2.2. (site descriptors) in a table. Why are these site descriptors not further discussed after the factorial analysis? Authors have found significant differences between sites/environments for all traits… Might the choice of other sites have led to different results?
  • Why are there no descriptives of the traits for each environment (as is done for each genotype (Table 2)? 
  • Blast resistance can, apart from the resistance genes in the rice plant, also depend on the pathogen type - which can vary between environments… Why did authors not study blast disease incidence (it is the first trait of interest);
  • Authors several times refer to "significant differences in GxE". What does that mean? As far as I know factorial analysis considers main effects of G and E as well as the significance of the interaction between G & E. If significant interaction prevails, the effect of G will depend on E and vice versa;
  • English style and grammar should be significantly improved - please consult a native speaker.

There are many other issues that I have highlighted in the annotated pdf and that authors should address.

Author Response

Reviewer 1 comments

Author response

1.                    

Description of the experiment should improve: it is e.g. not clear whether crosses were backcrossed with MR219 or Pongsu Seribu2.

Correction effected

Eighteen advance lines of BC2F2 generation from backcross of MR219 and Pongsu Seribu 2 were used in this study. Introgression lines were developed from these parents (MR219 x Pongsu Seribu 2) using marker-assisted backcross selection (MABS). The donor parent, Pongsu Seribu 2 (PS2), developed by Malaysian Agricultural Research and Development Institute (MARDI) possesses broad-spectrum resistance against blast fungal isolates. MR219 has high yielding potential with a suitable grain quality and good eating quality. Unfortunately, this variety is susceptible to blast. The F1 plant produced from the cross between MR219 and Pongsu Seribu 1 was later backcrossed with MR219 to produce BC1F1 and subsequently backcross again with MR219 to derive BC2F1. The BC2F1 was allowed to self to produce BC2F2 population among which eighteen advance lines having the highest genome recovery and phenotypic characteristics similar to the MR219 were selected [12]. The planting materials of 18 advance breeding genotypes and MR219 were subjected to multi-environmental field trials.

2.                    

Also the agronomic (authors call them cultural)practices should be described more in detail (e.g. fertilizer and pesticide type and dosage),

Correction effected

All the cultural practices from land preparation until harvesting were done following Malaysian Agricultural Research and Development Institute (MARDI) recommendation. The field was irrigated with an average of 10 cm water above the sea level throughout the experiments. Fertilization was applied following the recommendation from MARDI where 42 kg/ha muriate of potash and 57 kg/ha triple superphosphate were applied at day 15 after transplanting, 80 kg/ha of urea was applied in splits at 35, 55, and 75 days after transplanting. The pesticides; Malathion and Lambda-cyhalothrin (Warrior) were applied at the recommended rate when needed. Regular hand-weeding for narrow leaf was done to avoid interspecific competition among the plants. In the case of broadleaf, Southern Ag with active Ingredient halosulfuron methyl was applied.

3.                    

Number the formular of PVC, GVC, etc.

Correction effected

 i.               Phenotypic Coefficient of Variation (PCV)

Where:

 = phenotypic variance;

X= mean of the trait

ii.               Genotypic Coefficient of Variation (GCV)

Where;

 = genotypic variance;

X = mean of the trait

iii.               Broad Sense Heritability

Where:

 = genotypic variance;

 = phenotypic variance

iv.               Expected Genetic Advance

Where:

K = constant that represents the selection intensity (when k is 5% the value is 2.06);

= standard deviation of phenotypic variance;

 = heritability in a broad sense

4.                    

Discuss the correlation and cluster analysis in the material and methods section, not in the results

Correction effected

Correlation coefficients were analyzed using SAS Software (version 9.4) to study the relationship between traits. Multivariate analysis was done using SAS software to determine the pattern of interaction between the genotype and environment. The methods used in this study were principal component analysis (biplots) and cluster analysis. Biplot is an extensively used graphical method that displays the pattern of interactions among genotypes, environments and G×E interactions for identification of ideal genotype that are widely stable over environments or only fit for a specific environment. The Euclidean distances amidst the 19 improved genotypes evaluated were observed by using the standardized morphological data to construct an UPGMA dendrogram.

5.                    

Data on 13 quantitative rice traits were collected, that were not independent, as clearly shown from the significance of the correlation coefficients. Principal component analysis could have been performed prior to cluster analysis which might have given more insight in the phenotypic variability.

Correction effected following the reviewer’s comments.

6.                    

Cluster group III appears to contain the best (highest yielding) rice varieties but authors did not elaborate/conclude on how this finding helps breeders in selecting high-yielding blast resisting rice varieties for different environments.

Correction effected

7.                    

What other outstanding traits do genotypes in group III have that might assist breeders in selecting such varieties

Correction effected in the conclusion section

From the result obtained in this study, it is suffice to conclude that the genotypes in group III (G17 and G18) proved to be high yielding with 7.03 and 8.68 t/ha of grain yield, respectively. Group III clustered the genotypes that had high yield per hectare with good performances of other traits. In addition, their morphological performances were almost the same as the commercialized rice variety, MR219. The ability of these genotypes to maintain superiority in yield across the four environments showed that they have potentials for selection in further varietal improvement. Consequently, these genotypes can be fully utilized in a further breeding program for genotypes improvement. In particular, such genotypes would be useful to breeders in selecting high-yielding blast resisting rice varieties for different environments.  

8.                    

3.1. is a written description of what can be found in tables 1 and 2. Authors should find a way to shorten it.

Correction effected

9.                    

Summarize all data from 2.2 (site descriptors) in a table.

Correction effected

Table 1: Environmental data and description

Code

Planting period

Location

Altitude

(m)

Av. Temp.

Min – Max

(oC)

Av. Humidity (%)

Rainfall (mean)

EN1

Sept 2015 – Jan 2016

3° 25'N  101° 10'E

3

23- 31

83

782.4 (195.6)

EN2

Feb – Jun 2016

3° 25'N  101° 10'E

3

25- 37

65

482.7 (120.7)

EN3

Dec 2016 – March 2017

5° 59'N  100° 24'E

18

25 - 38

63

486.9 (121.7)

EN4

May – Sept 2017

3° 02'N 101° 42'E

32

24- 38

67

623.4  (115.9)

Note: EN1=Tanjung Karang, Selangor; EN2=Tanjung Karang, Selangor; EN3=Kota Sarang Semut, Kedah; EN4= Kota Sarang Semut

10.                 

Why are these site descriptors not further discussed after the factorial analysis?

Corrected accordingly. The manuscript has been revised following the reviewer’s suggestion.

11.                 

Authors have found significant differences between sites/environments for all traits…might the choice of other sites have led to different results?

The significant variation obtained between sites/environments for most traits is an important tool to the plant breeder. Without variation, the breeder has no work to do. However, stability analysis was conducted in the course of this research which showed that the genotypes were stable in the varied environments. Also cluster analysis was used to classify the genotypes into different groups and each cluster group varied from each other in terms of environmental factors. This is why we recommended that the cluster groups be used as varieties for commercial cultivation in various rice growing regions.

12.                 

Why are there no descriptive of the traits for each environment (as is done for each genotype (Table 2)?

Only results obtained on combined ANOVA for the four environments were presented for publication. We thought that presenting the results of the traits in each environment would make this article more cumbersome and difficult for the audience to read and comprehend.

13.                 

Blast resistance can, apart from the resistance genes in the rice plant, also depend on the pathogen type – which can vary between environments … why did authors not study blast disease incidence (it is the first trait of interest)

Blast disease incidence has been studied in our previous works and this particular research was targeted at assessing the agro-morphological performance, genetic parameters and clustering pattern in the listed environments. The lines assessed have been confirmed to be resistant to blast in the said environments.

14.                 

Authors several times refer to significant differences in GxE. What does that mean? As far as I know, factorial analysis considers main effects of G and E as well as the significance of the interaction between G & E. if significant interaction prevails, the effect of G will depend on E and vice versa.

Significant GxE interaction implies that the effect of G would depend on E and vice versa, as noted by the reviewer. This has been taken care of in the manuscript. Analysis was first conducted base on individual sites and later combined across four environments. For the sake of this publication, we presented tables for combined analysis.

15.                 

English style and grammar should be significantly improved – please consult a native speaker. There are many other issues I highlighted in the annotated pdf and that authors should address.

The manuscript has been revised following the reviewer’s comments. Some co-authors who are from native English speaking countries were assigned to improve the English style and grammar to avoid any use of English or grammatical error in the final version of this manuscript.

Reviewer 2 Report

The authors have clearly presented their study and its importance. I have a few specific comments:

-Lines 134-135: Were all traits normally distributed? If not, which traits were not normally distributed and was the phenotypic data transformed before further analysis?

-Lines 137-139 and Table 4: What data was used to estimate these correlations? Did you use the genotypic BLUPs or LS-Means or are these phenotypic correlations? In general, I would recommend reporting both phenotypic and genetic correlations. Because this is a well-balanced experimental design and all traits were measured in all environments, I would suggest using MANOVA to jointly analyse all 13 traits and estimate their generic correlations.

-Lines 244-247: Only correlations involving yield are mentioned. I would recommend discussing how the other traits were correlated with each other and whether the traits clustered together, similar to the genotypic cluster analysis. For example, Table 4 shows that traits that are phenologically similar are more highly correlated (e.g. NPH/NTH, FG/PFG/UFG, PH/PL).

Author Response

Reviewer 2 comments

1.        

The authors have clearly presented their study and its importance. I have a few specific comments

Thank you so much for your commendation.

2.        

Lines 134-135: were all traits normally distributed? If not, which traits were not normally distributed and was the phenotypic data transformed before further analysis?

It is a standard practice in our laboratory to do data transformation before data analysis using SAS software. All data collected on all traits were first subjected to square-root transformation before been analysed with SAS software.

3.        

Lines 137-139 and Table 4: what data was used to estimate these correlations? Did you use the genotypic BLUPs or LS-means or are these phenotypic correlations? In general I would recommend reporting both phenotypic and genetic correlations. Because this is a well-balanced experimental design and all traits were measured in all environments. I would suggest using MANOVA to jointly analyse all 13 traits and estimate their genetic correlations.

Multivariate analysis has been included following the reviewer’s suggestion.

4.        

Lines 244-247: only correlations involving yield are mentioned. I would recommend discussing how the other traits were correlated with each other and whether the traits clustered together, similar to the genotypic cluster analysis. For example, Table 4 shows that traits that are phonologically similar are more highly correlated (e.g. NPH/NTH, FG/PFG/UFG, PH/PL).

Correction effected following the reviewer’s suggestion.

Round 2

Reviewer 1 Report

The authors have made significant improvements to the paper and have addressed most of my comments appropriately. However, section 3.4. introduces methods that are not or insufficiently described in the Material and Methods section. What is GGE? What does “which-won-where” mean?  When does a genotype win in a sector? It is not clear what biplots represent. See e.g. Figure 1:  I have no idea how PCA-axes were determined or what the polygons and red lines (sectors?) mean or indicate. I wonder whether authors know what they were doing there other whether they just fail to provide a clear presentation of the method used (see lines 214-222). Also the abstract does not mention GGE biplot analysis. Why has GGE biplot analysi been added to the paper?

These unclarities make it very difficult to Judge the relevance of the findings or to assess the appropriateness of the discussion and conclusion.

Extensive editing of the english language is required. Authors claim to have engaged some native english speaking people. However these people most certainly did not address the newly written parts which are full of grammar and style mistakes.

Too many statements are unclear by inadequate choice of wording and/or lack of detail.

See more (minor) comments below

where is table 1?

lines

·        166-167: following my comment on the first version of this paper: authors reply that the 18 lines have been confirmed to resist blast in the four environments; but they should briefly mention it in the paper;

·        175-177: remove sentence. Just mention "(Table 1)".

·        178: ‘site description’ instead of ‘description’

·        193: recommendations

·        194: the sea level? please correct/clarify

·        197: how was 80 kg/ha of urea split? In three doses of 26.67 kg/ha?

·        198: replace ‘The pesticides; ’ by ‘The insecticides’;

·        198: remove ‘Warrior’

·        198: what are the recommended rates?

·        199: remove ‘to avoid interspecific competition among plants’

·        199-200: narrow leaf and broadleaf? You mean narrow leaf weeds and broadleaf weeds?

·        200: remove “Southern Ag with active Ingredient”; add dose;

·        214-222: not clear… add some references;

·        238: remove dot

·        242: were instead of was

·        244: combined

·        247: “GxE varied significantly” ; see my earlier remark… I would suggest to report interaction effects as “significant GxE interaction was observed…”

·        239-287: refer to the table where results are reported

·        239-287: I wonder if it is relevant to have this lengthy enumeration of all significant and insignificant effects, interaction effects and % of variation. All this information can straightforwardly be deduced from Table 2.

In the discussion and conclusion sections (line numbers start again from 1): lines

·        67: does Malaysia have a ‘fall’ period? If so, its months must be clearly defined. So why then write: “There is no doubt that the period falls within the Malaysian Fall.”?

·        68: not clear: varied? Differed from what?

·        73: add some references to underpin this statement

·        76: it is the variation in observed variables (not in genotypes)

·        79: replace ‘while’ by ‘however’

·        82: what varietal lines? Give some more detail.

·        81-84: yield traits? What traits… this is confusing; give more details.

·        136: remove: “From the result obtained in this study, it is suffice to conclude that the”

·        137: these are average yield figures? Of G, E or both?

·        136-140: please conclude instead of summarizing… where is this discussion of cluster contents leading us to?

Author Response

Thank you.
